# Residues of Persistent Organic Pollutants (POPs) in Agricultural Soils Adjacent to Historical Sources of Their Storage and Distribution—The Case Study of Azerbaijan

**DOI:** 10.3390/molecules25081815

**Published:** 2020-04-15

**Authors:** Aleksandra Ukalska-Jaruga, Karolina Lewińska, Elton Mammadov, Anna Karczewska, Bożena Smreczak, Agnieszka Medyńska-Juraszek

**Affiliations:** 1Department of Soil Science Erosion and Land Protection, Institute of Soil Science and Plant Cultivation–State Research Institute, Czartoryskich 8, 24-100 Puławy, Poland; bozenas@iung.pulawy.pl; 2Department of Soil Science and Remote Sensing of Soils, Adam Mickiewicz University in Poznan, Krygowskiego 10, 61-680 Poznan, Poland; karolina.lewinska@amu.edu.pl; 3Institute of Soil Science and Agrochemistry of Azerbaijan National Academy of Sciences, 5 M. Rahim, Baku AZ1073, Azerbaijan; elton.eldaroglu@gmail.com; 4Institute of Soil Science and Environmental Protection, Wroclaw University of Environmental and Life Sciences, ul. Grunwaldzka 53, 50-357 Wrocław, Poland; anna.karczewska@upwr.edu.pl (A.K.); agnieszka.medynska-juraszek@upwr.edu.pl (A.M.-J.)

**Keywords:** toxic substances, pesticides, polycyclic aromatic hydrocarbons, polychlorinated biphenyls, DDTs, HCHs, PCBs, PAHs, soil contamination

## Abstract

The aim of this study was to identify and examine the levels of organochlorine pesticides (OCPs), polycyclic aromatic hydrocarbons (PAHs), and polychlorinated biphenyls (PCBs) in soil collected from the surroundings of historical pesticide storage facilities on former agricultural aerodromes, warehouses, and pesticide distribution sites located in the most important agricultural regions in Azerbaijan. The conducted research included determination of three groups of POPs (occurring together), in the natural soil environment influenced for many years by abiotic and biotic factors that could have caused their transformations or decomposition. In this study, soil samples were collected in 21 georeferenced points located in the administrative area of Bilasuvar, Saatly, Sabirabad, Salyan and Jalilabad districts of Azerbaijan. Soil chemical analysis involved determination of organochlorine compounds (OCP): hexachlorocyclohexanes (HCHs) (three isomers α-HCH, β-HCH and γ-HCH) and dichlorodiphenyltrichloroethanes (DDTs) (six congeners 2,4′DDT; 4,4′DDT; 2,4′DDE; 4,4′DDE; 2,4′DDE; and 4,4′DDE); polycyclic aromatic hydrocarbons (PAHs): 16 compounds from the United States Environmental Protection Agency US EPA list and, PCBs (seven congeners identified with the following IUPAC numbers: 28, 52, 101, 118, 138, 153, and 180). Our research showed that OCPs reached the highest concentration in the studied areas. The total concentrations of OCPs ranged from 0.01 to 21,888 mg∙kg^−1^ with significantly higher concentrations of Σ6DDTs (0.01 μg kg^−1^ to 21880 mg kg^−1^) compared to ΣHCH (0.14 ng kg^−1^ to 166.72 µg kg^−1^). The total concentrations of PCBs in the studied soils was varied from 0.02 to 147.30 μg·kg^−1^ but only PCB138 and PCB180 were detected in all analyzed samples. The concentrations of Σ16 PAHs were also strongly diversified throughout the sampling areas and ranged from 0.15 to 16,026 mg kg^−1^. The obtained results confirmed that the agricultural soils of Azerbaijan contained much lower (up to by three orders of magnitude) concentrations of PCBs and PAHs than DDT. It is supported by the fact that PCBs and PAHs were not directly used by agriculture sector and their content results from secondary sources, such as combustion and various industrial processes. Moreover, the high concentrations of PAHs in studied soils were associated with their location in direct neighborhood of the airport, as well as with accumulation of contaminants from dispersed sources and long range transport. The high concentrations of pesticides confirm that deposition of parent OCPs have occurred from obsolete pesticide landfills.

## 1. Introduction

Persistent organic pollutants (POPs) have been recognized as substances that pose a serious risk to human health and the environment. Therefore, several groups of POPs including many individual compounds have been listed by the Stockholm Convention on Persistent Organic Pollutants [1]. The Stockholm Convention aimed at a range of actions focused on depletion and ultimately the elimination of POP releases into the environment, consequently limiting human exposure [1]. Because the Stockholm Convention is a “living document”, new groups of POPs had been added in all of the annexes over time, and they were also scheduled either for elimination or restricted production and use to reduce their releases from unintentional production. 

POPs represent a diverse group of different substances, which are toxic, semi-volatile, varied with mobility in the environment, and prone to long-range air transport on dust particles, accumulation in abiotic matrices, as well as bioaccumulation in living bodies [2,3]. This group of organic compounds, of natural or anthropogenic origin, possess a particular combination of physical and chemical properties such that, once released into the environment, they remain intact for exceptionally long periods of time, as they are resistant to photolytic, chemical and some of them on biological degradation [3,4,5,6,7]. Organochlorine pesticides (OCPs) as well as polychlorinated biphenyls (PCBs) and some polycyclic aromatic hydrocarbons (PAHs) belong to this group. The OCPs are organic molecules with linked chlorine atoms, high lipophilicity and, usually, high neurotoxicity [3,4,5,6,7,8]. Examples of OCPs are chlorinated insecticides, such as dichlorodiphenyltrichloroethane (DDT), aldrin, dieldrin, heptachlor, and endrin. OCPs are typically commercial synthetic organic compounds. In the 1940s, many chlorinated insecticides were widely applied due to their effectiveness in control of malaria and typhus as well as in agriculture and storage. However, in Europe and many countries world-wide use and production of OCPs are actually prohibited because of their severe environmental impact [8,9]. It was proved that e.g., in agriculture the efficiency of applied pesticides was very low due to their occurrence in powder form. In extreme cases, only 3% reached their targets but remaining part was spread out and could be found in the distance up to 2000 m away from the treated field [10]. Moreover, OCPs have been found in ground water reservoirs, seas and oceans. They show high stability and are subjected to accumulation in living organisms and sediments [4,11]. 

Polychlorinated biphenyls (PCBs) have been widely used as dielectric and coolant fluids in electrical converters, carbonless copy paper, and in heat transfer fluids, plasticizers in paints, plastics, and rubber products [3,12]. PCBs similar to OCPs undergo long air transport and have been found in areas far from their release sources [13]. As a consequence, they are found in some amounts in various matrixes all over the world. PCBs are still in use, although their manufacture has declined drastically since the 1960s, when a host of problems were identified. Some PCBs exhibit toxic effects to humans. They act as endocrine disruptors (notably blocking of thyroid system functioning), neurotoxins and carcinogens. 

Polycyclic aromatic hydrocarbons (PAHs) are formed as unintentional substances in all combustion and pyrolysis processes of organic materials [3,12]. PAHs are released to the environment from both natural and anthropogenic sources. Human activities (e.g., industry activity, emissions from individual house-heating systems, and road and air transport) currently are the main sources of PAHs, and, soil contamination with these compounds is especially serious problem in high-density industrial areas [5,14,15,16,17,18]. Therefore, the PAH emission profile may constitute an important diagnostic tool to identify their pollution sources [5,15,16,17,18]. The United States Environmental Protection Agency (US EPA) has listed 16 PAHs as priority pollutants since they are considered as either possible or probable human carcinogens [3,15,18]. Hence, their emission sources, distribution, content, and possible human exposure need more attention. 

POPs can be accumulated in different elements of the environment as a result of various natural and anthropogenic activities (pesticide application, emissions from industry and traffic, application of sewage sludge or compost, spills, and crop irrigation with contaminated water). POPs released to the atmosphere are a subjected to a long-range transport and soil-air exchange processes, but finally >90% of this compounds are deposited in the top soil horizon [11]. Soils are a significant sink for different groups of pollutants due to their high sorption capacity, porosity, and complexity. POP residues occurring in soils can adversely affect living organisms and plants and with contaminated food products enter the human food chain [19,20]. The fate and behavior of organic pollutants in soil is controlled by many different factors including soil characteristics (texture), chemical properties (pH, organic matter, and CEC) and environmental factors such as temperature and moisture. The extent to which POPs are susceptible in soils to different processes dependent on biodegradation efficiency and sorption capacity which determine their dissipation, resistance, and mobility [21].

The content of organic contaminant residues in soils is the net effect of biodegradation, leaching and/or volatilization, accumulation in soil biota as well as retention or/and sequestration by soil clay minerals and organic matter [22]. Maliszewska-Kordybach et al. [6,15] and Ukalska-Jaruga et al. [7,18] proved that the content of organic matter has a significant positive impact on the retention and persistence of POPs in soil. However, such relations were noticed for PAHs present in soils subjected to anthropogenic input. [5,6,7,15,18]. 

Due to the negative environmental impact of POPs, many research projects or monitoring programs at the national or regional level were carried out to determine the content of their residue in soils, especially from rural areas [23,24]. Several studies [4,5,6,16,19,23,24,25,26,27] characterized the distribution of pesticide residue in soil at the national or regional scale. Despite this, there is still a lack of data from many countries where POPs were emitted, distributed, stored and applied for many years. The aim of this study was to identify and examine the concentration levels of OCPs, PAHs, and PCBs in soil collected from the surroundings of historical pesticide storage facilities on former agricultural aerodromes, warehouses, and pesticide distribution sites located in the most important agricultural regions in Azerbaijan. The research included analysis of the residue of individual contaminants representing three POPs groups occurring together in the soils exposed to natural abiotic and biotic processes influencing POPs’ transformations for many years.

## 2. Results and Discussions

### 2.1. Assessment of OCPs Concentrations in Soils

Statistical results for the OCP concentrations in the soils are presented in Table 1. The total concentrations of OCPs (sum of HCHs and DDTs) ranged widely from 0.01 to 21,888.2 mg∙kg^−1^ (interquartile range from 19.5 to 1406.7 mg∙kg^−1^). The results indicated the sites with extremely high concentrations of the investigated compounds, which did not correspond with recent monitoring report of potentially contaminated sites, prepared for the purposes of the Stockholm Convention implementation in Azerbaijan (Azerbaijan Government Monitoring Group, 2005). The monitoring group of Azerbaijan for the UNEP Stockholm Convention analyzed the levels of OCPs and PCBs in soils of several obsolete pesticide stocks and suspected contaminated agricultural sites, and revealed that their median concentrations were in the ranges: 0.6–32.8 mg kg^−1^ (in obsolete pesticide sites) and 0.008–0.4 mg kg^−1^ (in agricultural areas). These results may confirm that deposition of parent OCPs have occurred from obsolete pesticide landfills. Moreover, higher concentration of OCPs and PCBs may result from the occurrence of hot spots—small areas exposed to high historical depositions. Avazova [27] examined OCP levels in soils in the agricultural areas of Azerbaijan during the period 1983–1992, with sampling campaigns organized twice in the same locations over a 10-year period. The research revealed that 12,600 ha of agricultural areas out of a studied 25,000 ha exhibited the Σ6DDTs (n = 50), at an average (range) concentrations of 0.573 (0.058–1.596) mg kg^−1^, and Σ3HCHs (n = 39) at concentrations of 0.028 (0.004–0.086) mg kg^−1^. The application rates of DDT in the agricultural areas in the 1980s, according to Avazova [27] were in the range 0.1–1.12 kg ha^−1^ (expressed as active pesticide substance). 

Among all OCPs, the dominant group, present in soils in the highest concentrations, were 4,4′DDT and their isomers (2,4′DDE, 4,4′DDE, 2,4′DDD, 4,4′DDD, and 2,4′DDT)—Table 1. The average content of ∑6DDT compounds ranged from 5.6 × 10^−3^ mg kg^−1^ to 21,885.9 mg kg^−1^ (interquartile range: 15.5–1395.4 mg kg^−1^), and constituted 98.5% of all determined pesticides. The median concentrations of individual compounds were 2.2 mg kg^−1^, 1.7 mg kg^−1^, 0.6 mg kg^−1^, 1.2 mg kg^−1^, 8.7 mg kg^−1^, 25.4 mg kg^−1^, respectively, for 2,4′DDE, 4,4′DDE, 2,4′DDD, 4,4′DDD, and 2,4′DDT, and 4,4′DDT. Those values were considerably higher compared to similar reports published by other investigators, related to various European countries [4,6,7,19,25,26,27]. As mentioned before, the data reported by Aliyeva [28] from Azerbaijan were much lower than our results, but these studies were carried out in other area of Azerbaijan. According to Li et al. [29], the total amount of DDTs used in the Azerbaijan in the years 1946–1990 was assessed as 250,000–520,000 tones. The main source of DDT in environmental samples in Europe and Asia was a commercially distributed product called Polydofen-60 (liquid mixture, contained 20% DDT and 40% Toxaphene as active substances). Its total production in FSU was 44,000 tones, but there is a lack of information about its usage in Azerbaijan. Although 4,4’DDT is chemically and biochemically stable with an estimated half-life time in soil is about 3–10 years (PPBD: Pesticides Properties Database), in our study this compound was responsible for 2.3%–64.0% of total Σ6DDT concentrations in soils. The isomeric composition of DDT can be influenced substantially by different environmental processes. The abundance of 2,4’-DDE and 4,4’-DDE in samples indicates their environmental degradation in the processes of weathering and aging [28,30], which may be related to microbial activity, sorption by organic matter or the influence of other environmental factors. The abundance of 2,4′-DDE and 4,4’-DDE metabolites in soils were overall responsible for 10% of the total ∑6DDTs content. The ratio of 4,4’DDD to sum of 4,4’DD D + 4,4’DDE + 4,4’DDT varied from 0 to 71% what indicate on mixed pollution history or limited 4,4’DDT and 2,4’DDT biological transformations in soils. DDT present in soil may undergo aerobic or anaerobic transformations. Under aerobic conditions, DDT is degraded to DDE while under anaerobic circumstances DDD predominates. High percentage of 4,4′-DDE confirms on microbial degradation in presence of oxygen in soils. The ratio of 4,4′-DDT/(4,4′-DDE + 4,4′-DDD) is an indicator whether 4,4′-DDT in soils is degraded or inputted recently [27,31]. In the present study, the proportion of 4,4′-DDE/4,4′DDD comparatively accounted for 0.9–6.46 and the ratio of 4,4′-DDT/(4,4′-DDE + 4,4′-DDD) ranged from 0.41 to 61.92 with a mean value of 8.76. The results suggested that the DDT residues are mainly subjected to anaerobic degradation in most sampling sites, but generally high levels of DDTs concentration can affect difficult degradation of these compounds. Moreover, the 2,4′-DDT/4,4′-DDT ratio can be used to distinguish DDT sources [31]. The 2,4′-DDT/4,4′-DDT ratio varies from 0.02 to 12.42 with mean value of 0.27 indicating that the DDT residues derived from technical DDT input and dicofol. The 2,4′-DDT/4,4′-DDT ratio for 48% of samples exhibited value above 0.3, implying the dominant contribution of mixed sources of DDT [31].

The concentrations of Σ3HCHs were much lower than those observed for DDTs and ranged from 0.1 × 10^−4^ µg kg^−1^ to 166.7 µg kg^−1^. Alyieva et al. [28] reported lower concentrations of HCH in soils in Azerbaijan, varying in the range 0.9 × 24.5 µg kg^−1^. The dominant compound was α-HCH, which contributed about 73 % of the Σ3HCHs, followed by β- HCH and γ-HCH (Table 1). The isomer ratio of α-HCH to γ-HCH for most samples significantly exceeded 1.0, which confirms the high stability of the α isomer, while γ-HCH is subject to fast biological and chemical degradation and indicates ability to leaching because of its relatively high water solubility [7,19,32]. The α-HCH and γ-HCH can both be converted to β-HCH, which exhibited the lowest degradation rate and vapor pressure [6,31,32]. In our study, the proportion between this isomers (β-HCH / (α-HCH + γ-HCH)) were from 0.0 to 5.0 with median value of 2.1, which points out on a historical nature of these contaminants in soils or its deposition caused by long-range transport [6,31].

The differences in composition of HCH isomers in the environment could indicate different sources of contamination. Technical HCHs contain high concentrations of α-HCH (60%–70%), compared to β-HCH (5%–12%) and γ-HCH (10%–12%). Therefore, despite the application of technical HCHs which has been forbidden in many years ago, its long degradation time may cause the high persistence of these contaminants in soil [31]. 

### 2.2. The PCB Concentration Level in Soils 

The total concentrations of PCBs in soils varied within a wide range from 0.02 to 147.3 μg·kg^−1^ with a median of 0.06 μg kg^−1^. Among determined compounds, only PCB138 and PCB180 were detected in all samples (Table 2). The PCB pattern, characterized in terms of individual homologs, indicated a dominant content of six-chlorinated congeners, which content accounted for 20%–100% of total PCB concentrations. The obtained results confirmed that the agricultural soils contained much lower concentrations of PCBs (up to three orders of magnitude) than urban soils, what is supported by the fact that PCBs were not directly used in agriculture. The cities and industrial areas are generally considered to be important emission sources of PCB [25]. According to International Pollutants Elimination Network, the largest part of PCB deposition in Azerbaijan (reported in 2019) is associated with the unsecured storage of 83 tons of chemicals, including approximately 34.5 tons of PCBs oils that need to be utilized [33]. As so far, no arrangements for storage of these materials were found, suggesting a lack of control over circulation of PCBs-containing equipment; and their possible emission into the environment [33]. Moreover, PCBs can evaporate from secondary sources, such as contaminated buildings, electricity equipment, combustion processes, and various industrial activities. In the ambient environment, PCB congeners partition into the gaseous and particle phases [13,34]. Generally, low chlorinated congeners are more susceptible to long-range atmospheric transport to areas outside their source site than the higher chlorinated congeners [13,34,35]. So, the patterns of PCB homologs observed in this study may have originated from the short range atmospheric transport from industrial sites, and medium-range regional atmospheric deposition. Moreover, in soils subjected to contaminants deposition caused by long-range transport are predominantly scavenging by vegetation before they accumulate in the soil [25,34].

The reported concentrations of PCBs in Azerbaijan soils exhibited higher values than those found in India [35], China [25], Portugal [13], and Iran [36], but significantly lower compared to those from USA [34,37,38].

### 2.3. The PAHs Concentration Level in Soils

The concentrations of all 16 PAHs were strongly diversified throughout the sampling areas (Table 3), and ranged from 0.15 to 16,086.7 mg kg^−1^. The highest average concentrations were characteristic for four-ring compounds, such as: Chry (391.8 mg kg^−1^), Fla (214.3 mg kg^−1^), Pyr (141.3 mg kg^−1^), and BaA (16.2 mg kg^−1^), while the sum of two ring (Nap) and three ring (Acy, Ace, Fl, Phe, and Ant) PAH compounds, as well as five ring (Bbf, BkF, and BaP) and six ring (IcdP, DahA, and BghiP) compounds were present at much lower concentrations, with the average values of 4.1 mg kg^−1^ and 2.1 mg kg^−1^, respectively. Despite the very high average concentrations of some individual PAHs, the median value of Σ16PAHs was generally low, at the level of 0.7 mg kg^−1^, which still gives a much higher content than that recorded from China [39], Japan [40] or European countries, such as Norway [41], Poland [15,18,42,43], and Czech Republic [19]. 

Overall, the medium-molecular hydrocarbons (four ring compounds) contributed more pronouncedly to the total content of PAHs than the lower (two and three ring hydrocarbons) and high (five and six ring hydrocarbons) molecular ones; the groups of four rings accounted for 99.5%, while low and high-molecular PAH compounds accounted for 0.36% and 0.1% of the Σ16PAH concentrations, respectively. High variability of PAH content in the examined sites indicate the occurrence of ‘hot spots’, i.e., local points with particularly high concentrations of harmful compounds. Moreover, the high concentrations of PAHs in studied soils were associated with their location in direct neighborhood of the airport, as well as with accumulation of contaminants from dispersed sources and probably long range transport. According to the International Pollutants Elimination Network data [33] estimated emission load of PAHs in Azerbaijan amounts to 4207 kg y^−1^ for BaP, 5548 kg y^−1^ for BbF and 1918 kg y^−1^ for BkF.

According to the criteria described in the publication of Maliszewska-Kordybach [44] the most of the soils analyzed in our study can be classified as non-contaminated (10% of samples) and weakly contaminated (52% of samples) because they were characterized by the ∑16PAHs concentration below 200 µg kg^−1^ and 200–600 µg kg^−1^, respectively. The rest of the soils can be recognized as contaminated (10% of samples with PAHs concentration between 600 and 1000 µg kg^−1^) and heavily contaminated (29% of samples with PAHs concentration below 1000 µg kg^−1^). Maliszewska-Kordybach [44] proposed four classes of soil contamination by PAH. The threshold values express the absolute sum of the 16 PAH concentration in the soil, disregarded PAH composition and soil properties. The limit values at the level 200, 600 and 1000 µg kg^−1^ have been derived from the results of determinations of PAH content in European soils [44].

The majority of PAHs in agricultural regions originated from anthropogenic sources, such as coal and wood burning, petrol and diesel oil combustion and industrial processes [15,17,18,43]. The PAH diagnostic ratios of selected individual compounds may provide an important tool for the identification of their pollution sources, and the PAH emission profile. During pyrogenic processes (including wood burning and coal burning), low-molecular-weight PAHs are usually formed, whereas petrogenic processes, such as the combustion of fuels in engines, emit higher-molecular-weight PAH compounds [16,17]. Isomeric relationships of individual compounds are used as markers in identifying the sources of their origin. For this purpose, Maliszewska-Kordybach et al. [5] applied proportions between Fla/(Fla + Pyr) while Akyüz and Çabuk [45] and Klimkowicz-Pawlas et al. [43] proposed BaA/(BaA + Chr) ratios. In both cases, the calculated indices pointed on a petrogenic source of soil contamination, because their values were below 0.5 and above 0.35, respectively, for Fla/(Fla + Pyr) and BaA/(BaA + Chr) ratios. Our results indicate that the majority of identified PAHs originated from petrogenic sources (Figure 1). Therefore, the mutual proportions between individual PAHs confirms that car transport was their main emission source. 

### 2.4. Mutual Relations between Analyzed POPs

The mutual relationship between the analyzed compounds may indicate similarities or differences in the processes of their transformation in soil. The factor analysis based on principal component analysis (PCA) method was applied for this kind of investigations. This approach allowed for a detailed analysis of the dependences between individual group of POPs which are unnoticeable by simple Pearson’s correlations. The PCA was performed with varimax rotation. The total data variations within the data set were exemplified by three PCA components that explained almost 83% total data variability (PCA 1: 32%, PCA 2: 30%, and PCA 3: 20%) (Table 4). The first factor exhibited high significant correlation (r > 0.96) with Σ6DDT and Σ7PCB which indicates that transformations and perhaps the way and time of their accumulation in the soil were convergent. Similarly, the second factor was strongly related with concentrations of two and three-ring PAHs (r > 0.93) as well as five and six ring PAHs (r > 0.93) in soils which may be assigned to the co-commission and deposition processes of PAHs. Admittedly, molecular index analyses have indicated the origin of PAHs from petrogenic sources, however Tobiszewski and Namiesnik [17] indicated that organic compounds may be cracked to reactive radicals, which react to form stable PAHs. These PAHs are less alkylated, and their molecules contain more aromatic rings than petrogenic low molecular weight PAHs [16,17]. A third factor, which was significantly correlated (r > 0.73) with medium molecular PAHs and Σ3HCH, probably resulted from their highest concentrations in the studied soils. The PCA analysis showed the mutual relationships between compounds that affect their content during aging. According to Ehlers and Loibner [46], ageing is an important determinant of the sorption characteristics of organic contaminants in the soil under long-term anthropopressure. Moreover, Gao et al. [47] studied the ageing effects on mechanism-specific sorption and desorption which showed that ageing time leads to higher retention, reduction of desorption and lessens extractability as a result of the greater contaminant sequestration or binding. Factors of PCA analysis indicate the possible similarities in the aging processes between the studied contaminants. 

Unfortunately, all the sampling sites included in this study are currently used as agricultural lands such as pasture, cornfield, alfalfa field and private gardening with extensive irrigation and stream or groundwater. Despite the fact that many years passed, OCPs still persist in some sites on soil surface, as well as in the areas of aerodromes and warehouses, which were destroyed and often abandoned. Such sites or larger areas pose a huge threat to the environment and, more importantly, to human beings. For this reason, they should be monitored and properly secured until the time when necessary funds will be obtained for their remediation.

## 3. Materials and Methods 

### 3.1. Site Description and Sample Collection

Soil samples were collected from 21 georeferenced points located in the administrative area of Bilasuvar, Saatly, Sabirabad, Salyan and Jalilabad districts of Azerbaijan (Figure 2), which were known for a long cultivation history of cotton, grapes and cereals in those regions. Those samples represent sites, where pesticides were distributed, stored and reloaded at the aerodromes (Table 5). 

The climate in the region is arid, with average air temperature of 14 °C and average annual precipitation of 200–250 mm. Rainfalls occur mostly from November to May. The spatial distribution of soil units is highly related to the geological, geomorphological and climatic features of the region. According to the national soil classification and its correlation to the international standard for soil classification (WRB) [48], accommodating soil groups are Calcisols and Kastanozems with aridic and xeric pedoclimatic regimes, respectively. Water table in the region is rather high, depending on the season it varies between 0.5–2.5 m. The soils of the whole region are mostly salt-affected due to irrigation, and indicate various types and rates of salinization. 

Due to the long agricultural history of this region, the large amounts of pesticides were used for years. The rates of pesticide application were enormously high. According to Fedorov and Yablokov [10], the annual average doses of pesticides used in Azerbaijan, in the 1980s, were up to 237 kg ha^-1^. Additionally, one of the five largest factories that produced pesticides in Soviet Union was located in Azerbaijan (the Sumgait factory, located 35 km from the capital, Baku). The factory manufactured and stored various pesticides, including DDT (dichlorodiphenyltrichloroethane) and HCH (hexachlorocyclohexane) [29]. From 1951 to 1980, the Sumgait factory produced 480.5 tons of 5% DDTs and about 30,449 tons of HCHs [49] that were widely applied across the former Federal Soviet Union (FSU). It is estimated that in one of pesticide cemetery, located in Gobustan region, gathered approx. 8000 tons of banned and obsolete pesticides [50], but unprotected old pesticides warehouses and local distribution sites can still be found in many places. 

Our study was focused on places like warehouses (2 sampling sites), aerodromes (4 sampling sites) and pesticide distribution sites (8 sampling sites), where OCPs were stored. The most serious pollution, visible to the naked eye and easy to recognize due to bad smelling, occurs in the surroundings of airstrips. Numerous warehouses, densely distributed across the administrative regions, corresponding to the number of collective farming systems, were abandoned and destroyed when the FSU collapsed, leaving obsolete pesticides in unprotected containers. Some of aerodromes and warehouses have been completely destroyed, leaving some remnants such as the patches of concrete in former airstrips or the ruins of old warehouses. In all the sites included in the study, DDT and HCHs were used as diluted with water and sprinkled by tractor-carried sprayers without using special protective equipment. Selected locations were potentially also affected by other POP groups: PAHs and PCBs. PAHs might have originated form exhausts of airplanes, tractors and car transporting pesticides emissions as well as released from factories and domestic cheating sources using various sources of organic matter. PCBs might have occurred form unprotected, destroyed electricity equipment.

Soil samples were collected from the sites suspected to indicate particularly strong contamination with POPs. In aerodromes, the samples were collected along the airstrips. In the old warehouses, soil sampling sites were appointed in those places where pollution was particularly likely, which was identified by odor secretion or by barren land surface, without plant cover. The sampling areas were selected from the Pure Earth Organization database (https://www.pureearth.org/), and the soil samples were collected from a surface soil layer (0–15 cm depth) following guidelines offered by Pure Earth Organization [49] in February, 2014. Plants, plant residues, and stones were removed prior to sample storage and analysis. 

### 3.2. POPs Analysis

Chemical analysis involved determination of organochlorine compounds and polycyclic aromatic hydrocarbons in soil samples. Organochlorine compounds analysis included three groups, PCBs, HCHs and DDTs. The PCBs included seven congeners identified with the following IUPAC numbers: 28 (2,4,4′-tichlorobiphenyl, 3-chlorinated PCB), 52 (5,5′-tetrachlorobiphenyl, 4-chlorinated PCB), 101 (2,2′,4,5,5′-pentachlorobiphenyl, 5-chlorinated PCB), 118 (2,3′,4,4′,5 pentachlorobiphenyl, 5-chlorinated PCB), 138 (2,2′,3,4,4′,5′-hexachlorobiphenyl, 6-chlorinated PCB), 153 (2,2′,4,4′,5,5′-hexachlorobiphenyl, 6-chlorinated PCB) and 180 (2,2′,3,4,4′,5,5′-heptachlorobiphenyl, 7-chlorinated PCB), as specified by the Institute for Reference Materials and Measurements. Furthermore, three HCH congeners: α-HCH (α-Hexachlorocyclohexane), β-HCH (β-Hexachlorocyclohexane), and γ-HCH (γ-Hexachlorocyclohexane), and six DDT compounds, namely: 2,4′DDT (1,1,1-Trichloro-2-(2-chlorophenyl)-2-(4-chlorophenyl)ethane), 4,4′DDT (1,1,1-Trichloro-2,2-bis(4-chlorophenyl)ethane), 2,4′DDE (2-(2-Chlorophenyl)-2-(4-chlorophenyl)-1,1-dichloroethene), 4,4′DDE (1,1-Dichloro-2,2-bis(4-chlorophenyl)ethene), 2,4′DDE (2-(2-Chlorophenyl)-2-(4-chlorophenyl)-1,1-dichloroethene) and 4,4′DDE (1,1-Dichloro-2,2-bis(4-chlorophenyl)ethene) were included in the studies. All organochlorine compounds were extracted and determined during one analytical procedure fully described by Maliszewska-Kordybach et al. [6,9]. Quality control of analytical procedure comprised sample blank check with each analytical series, spiked quartz as laboratory check standard and duplicate samples in every 20 samples set. The MDLs were between 0.02–0.05 µg·kg^−1^, 0.03–0.17 µg·kg^−1^ and 0.03–0.06 µg·kg^−1^ while recoveries were in the range 82%–92%, 71%–82% and, 59%–67% for PCBs, DDTs, and HCHs, respectively. 

Determination of PCB, DDTs and HCHs was performed in a single run using gas chromatography with microelectron capture detection, and for selected samples was additionally confirmed by gas chromatography-triple mass spectrometry in selected ion monitoring mode with diagnostic ions, as described below for PAHs. The gas chromatography-microelectron capture detection system was an Agilent 6890 (Agilent Tech., Santa Clara, CA, USA), gas chromatograph equipped with a ^63^Ni microelectron capture detector and fused-silica capillary column (DB-5, 30 m × 0.32 mm I.D., 0.25 μm film thickness, Agilent Technologies). 

PAH analysis methodology was fully described by Maliszewska-Kordybach et al. [5] and Ukalska-Jaruga et al. [18]. PAHs included 16 compounds according to US EPA List: Naphthalene-Nap, Acenaphthylene-Acy, Acenaphthene-Ace, Fluorene-Fl, Phenanthrene-Phe, Anthracene-Ant, Fluoranthene-Fla, Pyrene-Pyr, Benzo(a)anthracene-BaA, Chrysene-Chr, Benzo(b)fluoranthene-Bbf, Benzo(k)fluoranthene-BkF, Benzo(a)pyrene-BaP, Indeno(1,2,3-cd)pyrene-IcdP, Dibenz(a, h)anthracene-DahA, and Benzo(ghi)perylene-BghiP. The analytical procedure included extraction of ground samples (grain size ≤ 0.10 mm) with dichloromethane in a ASE200 Accelerated Solvent Extractor (Dionex Co., Sunnyvale, CA, USA). Before extraction, the ground samples were mixed with 2 g of diatomaceous earth and spiked with 10 μL of a recovery standard containing five deuterated PAHs (PAH-31, Dr. Ehrenstorfer GmbH, Augsburg, Germany). Clean-up of the concentrated extract was performed using hexane elution on activated silica gel (16 h at 135 °C). The concentrations of the 16 US EPA PAHs were determined by gas chromatography triple mass spectrometry on an Agilent 7890B GC system (Agilent Tech., Santa Clara, CA, USA), equipped with an Agilent 7000C detector and Agilent 7693 Autosampler. Sample analysis was performed in multiple reactor monitoring (MRM) mode with diagnostic ions as recommended in ISO 22892:2006. Quality control measures included analysis of a certified reference material (CRM 131, ANAB Accredited Tested Laboratory), duplicate matrix samples and a solvent blank sample. The precision expressed as a relative standard deviation (RSD) was in the range of 5%–12% and the recovery for individual compounds from CRM 131 was within 62%–84%. The limit of quantification (LoQ) for individual PAH compounds ranged from 0.02–2.10 µg kg^−1^, while the limit of detection (LoD) fitted within the 0.01–0.81 µg kg^−1^ range.

### 3.3. Statistical Analysis

The Statistica software (Dell Statistica, version 13.3, TIBCO Software Inc., Greenwood Village, CO, USA) was used for statistical analysis. Basic statistical parameters such as mean, median, extreme values (min, max), lower quartile (LQ), upper quartile (UQ) and coefficient of variation (CoV) were calculated. The chi-squared test was applied for checking the normal distribution of the results, and the principal component analysis (PCA) was used to provide an overview of the relations between individual groups of analyzed pollutants. PCA used an orthogonal transformation to convert a set of observations of possibly correlated variables into a set of values of linearly uncorrelated variables called principal components. The number of significant PCA factors was determined based on the screen plot. 

## 4. Conclusions

Persistent organic pollutants represent a diverse group of toxic substances, which are prone to long-range transport and have a high susceptibility to bioaccumulate in tissue of living organism, therefore the assessment of their concentrations is very important for environment protection and human health. The research conducted in Azerbaijan has shown that the agricultural soils of this area contained high concentration of DDT and significantly lower PCB and PAH contents. 

It is supported by the fact that PCBs and PAHs occurrence in soils result from secondary sources, such as combustion and various industrial processes. Moreover, sampling sites were located in direct neighborhood of the airport, transport roads, and were exposed to PAHs originated from dispersed sources and atmospheric deposition. The high concentrations of pesticides confirm that parent OCPs have occurred from obsolete pesticide landfills. Moreover, the high OCP content in soils may be related with the appearance of hot spots—small areas exposed to high historical depositions resulting from the location of pesticide distribution points. 

## Figures and Tables

**Figure 1 molecules-25-01815-f001:**
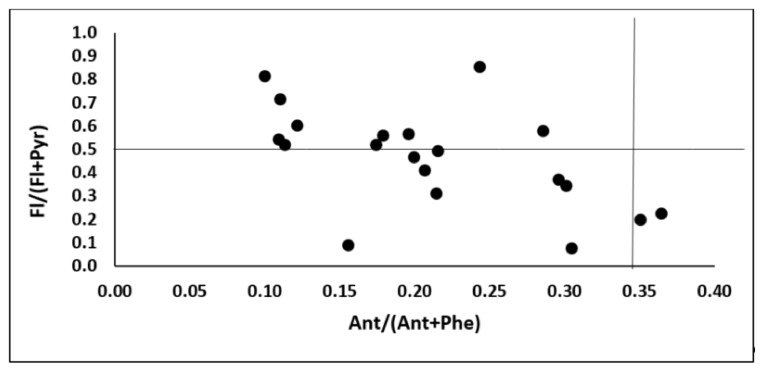
The PAHs diagnostic ratios for identification the source of soil pollution.

**Figure 2 molecules-25-01815-f002:**
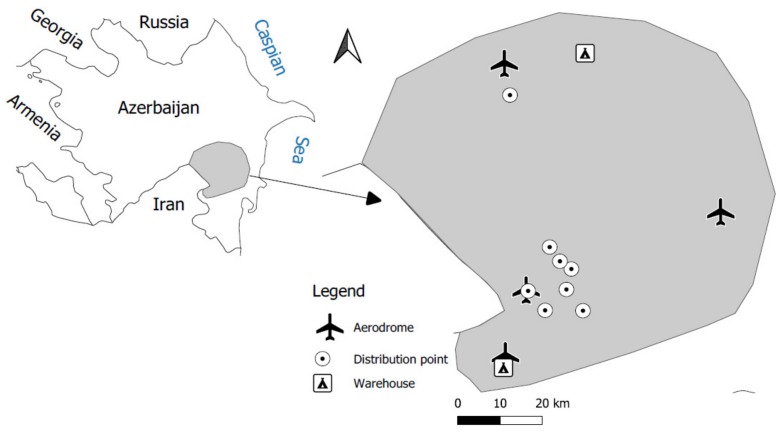
Scheme of the sampling area containing designated warehouses (2 sampling sites), aerodromes (4 sampling sites), and pesticide distribution sites (8 sampling sites).

**Table 1 molecules-25-01815-t001:** Statistical evaluation of organochlorine pesticides (OCPs) concentrations in soil (n = 21).

Compound.	Min.	Max.	Average	Me	Lower Q	Upper Q	SD	CoV %
	µg kg^−1^							
αHCH	< d.l.	141.1	14.0	3.9	0.1	10.9	31.2	222
βHCH	< d.l.	21.4	3.3	1.2	0.0	2.6	6.0	182
γHCH	0.1 × 10^−4^	12.8	1.9	0.5	0.2	1.7	3.2	167
∑3HCH	< d.l.	166.7	19.3	4.6	2.2	25.3	36.8	191
	mg kg^−1^							
2,4′DDE	0.3∙10^−3^	131.7	18.9	2.2	0.4	34.7	32.9	175
4,4′DDE	< d.l.	398.1	63.8	1.7	0.3	32.8	122.3	192
2,4′DDD	< d.l.	157.9	23.2	0.6	0.0	4.5	44.3	191
4,4′DDD	< d.l.	1207.9	73.9	1.2	0.6	3.1	264.6	358
2,4′DDT	0.2 × 10^−3^	3933.9	392.4	8.7	3.3	286.6	915.8	233
4,4′DDT	4.7 × 10^−3^	16311.3	1478.6	25.4	11.9	956.3	3703.3	250
∑6DDT	5.6 × 10^−3^	21885.9	2050.7	38.9	15.5	1395.4	4998.6	244
∑OCP	0.01	21888.2	2067.0	43.1	19.49	1406.0	5001.4	241

n.d.—not defined, Min.—the lowest content of the test compound, above the detection limit, Max—the highest content of the analyzed compounds, Me—median, Lower Q—lower quartile, Upper Q—upper quartile, SD—standard deviation, CoV—coefficient of variation, n—number of samples, and < d.l.—below detection limit.

**Table 2 molecules-25-01815-t002:** Statistical evaluation of polychlorinated biphenyls (PCBs) concentrations in soils (n = 21).

	Min.	Max.	Average	Me	Lower Q	Upper Q	SD	CoV %
	µg kg^−1^							
PCB 28	n.d.	n.d.	n.d.	n.d.	n.d.	n.d.	n.d.	n.d.
PCB 52	n.d.	n.d.	n.d.	n.d.	n.d.	n.d.	n.d.	n.d.
PCB 101	n.d.	n.d.	n.d.	n.d.	n.d.	n.d.	n.d.	n.d.
PCB 118	n.d.	n.d.	n.d.	n.d.	n.d.	n.d.	n.d.	n.d.
PCB 138	0.01	0.41	0.05	0.05	*n.d.*	0.03	0.10	206
PCB 153	n.d.	n.d.	n.d.	n.d.	n.d.	n.d.	n.d.	n.d.
PCB 180	0.01	147.30	8.91	0.02	0.01	0.38	32.04	359
∑PCB	0.02	147.30	3.57	0.06	0.01	0.60	31.99	357

n.d.—not defined, Min.—the lowest content of the test compound, above the detection limit, Max—the highest content of the analyzed compounds, Me—median, Lower Q—lower quartile, Upper Q—upper quartile, SD—standard deviation, CoV—coefficient of variation, and n—number of samples.

**Table 3 molecules-25-01815-t003:** Statistical evaluation of polycyclic aromatic hydrocarbons (PAHs) concentrations in soils (n = 21).

	Min.	Max.	Average	Me	Lower Q	Upper Q	SD	CoV %
	mg kg^−1^							
Nap	0.01	16.9	0.83	0.02	1.0∙10^−2^	0.025	3.7	442
Acy	< d.l.	0.3	0.03	0.01	0.2∙10^−2^	0.007	0.1	265
Ace	< d.l.	20.7	1.01	0.01	0.2∙10^−2^	0.013	4.5	446
Fl	< d.l.	7.8	0.39	0.01	0.5∙10^−2^	0.021	1.7	434
Phe	< d.l.	4.5	0.28	0.06	4.5∙10^−2^	0.118	1.0	340
Ant	0.01	2.7	0.19	0.02	1.2∙10^−2^	0.026	0.6	329
Fla	< d.l.	4498.3	214.3	0.03	1.9∙10^−2^	0.091	981.5	458
Pyr	0.01	2964.2	141.3	0.04	2.9∙10^−2^	0.132	646.7	458
BaA	< d.l.	337.9	16.2	0.01	0.2∙10^−2^	0.048	73.7	456
Chr	< d.l.	8226.1	391.8	0.04	3.0∙10^−2^	0.105	1794.9	458
BbF	< d.l.	2.1	0.26	0.04	1.0∙10^−2^	0.147	0.5	205
BkF	< d.l.	1.6	0.15	0.01	0.3∙10^−2^	0.071	0.4	233
BaP	< d.l.	2.4	0.19	0.03	0.2∙10^−2^	0.058	0.5	292
IcdP	< d.l.	1.0	0.11	0.02	0.6∙10^−2^	0.054	0.2	225
DahA	< d.l.	0.2	0.03	0.01	0.3∙10^−2^	0.024	0.1	169
BghiP	< d.l.	0.9	0.09	0.01	0.2∙10^−2^	0.058	0.2	242
2 + 3-ring PAHs	0.04	50.9	2.7	0.12	10.2∙10^−2^	0.2	11.1	404
4-ring PAHs	0.04	16,025.3	763.6	0.13	9.3∙10^−2^	0.6	3496.9	457
5 + 6-ring PAHs	0.01	6.9	0.8	0.12	4.2∙10^−2^	0.5	1.8	213
∑16PAH	0.15	16,086.7	4.6	0.67	2.4∙10^−2^	1.5	3496.3	455

d.l.—detection limit, Min.—the lowest content of the test compound, above the detection limit, Max—the highest content of the analyzed compounds, Me—median, Lower Q—lower quartile, Upper Q—upper quartile, SD—standard deviation, CoV—coefficient of variation, n—number of samples, and < d.l.—below detection limit.

**Table 4 molecules-25-01815-t004:** Factor analysis, factor loading matrix correlation used to generate the principal component analysis (PCA components (after varimax rotation).

	PCA 1	PCA 2	PCA 3
2 + 3P WWA	−0.11	0.92 *	0.16
4P WWA	0.23	−0.14	0.73 *
5 + 6P WWA	−0.15	0.93 *	0.13
∑7PCB	0.96 *	0.11	−0.20
∑3HCH	0.15	−0.17	0.75 *
∑6DDT	0.96 *	0.19	−0.06
% of variance	33	30	20
Cumulative %	33	63	83

* Significant correlation (*p* > 0.05).

**Table 5 molecules-25-01815-t005:** Description of the sampling points localization (n = 21).

Name of Sampling Area	Administrative District	Area, Hectare	Type	Land Use	No. of Samples
Jurali	Bilasuvar	0.21	Distribution site	Pasture	1
Amankend	Bilasuvar	0.15	Distribution site	Pasture	1
Aribatan	Salyan	11.39	Aerodrome	Pasture	2
Khirmandali	Bilasuvar	1.19	Aerodrome	Arable	2
Agalykend	Bilasuvar	0.02	Distribution site	Settlement	1
Yukhary Agaly	Bilasuvar	0.59	Distribution site	Arable	2
Garatapa	Sabirabad	5.85	Warehouse	Pasture	2
Abazalli	Jalilabad	0.39	Aerodrome	Pasture	2
Gunashli	Bilasuvar	0.17	Distribution site	Settlement	2
Takla	Jalilabad	0.09	Warehouse	Settlement	2
Dadagorgud	Saatly	0.38	Distribution site	Pasture	1
Molday	Saatly	0.25	Aerodrome	Pasture	1
Zahmatabad	Bilasuvar	0.15	Distribution site	Pasture	1
Chaily	Bilasuvar	0.23	Distribution site	Pasture	1

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
