# Peer review of "Residues of Persistent Organic Pollutants (POPs) in Agricultural Soils Adjacent to Historical Sources of Their Storage and Distribution—The Case Study of Azerbaijan"

_molecules, 2020, doi:10.3390/molecules25081815_

Round 1

Reviewer 1 Report

Title: Residues of persistent organic pollutants (POPs) in agricultural soils adjacent to historical sources of their storage and distribution. The case study of Azerbaijan.

This paper deals with a very interesting topic, because it takes into consideration an important problem like residues of Persistent Organic Pollutants (POPs) in agricultural soils in Azerbaijan, a Country where in the past many tons of contaminants were used. I agree with the authors when they say that it is very important to know POPs amount and to residue it because of human and environmental health risks associated with their existence.

The paper is well structured, well done and easy to read. All necessary information is reported, so the paper is coherent.

This paper collected more information and it can be useful as the starting point for other studies as for example human biomonitoring studies. Do you know if human biomonitoring studies were carried out in the same regions where you picked up your samples? It would be very interesting to have POPs levels in women and men (also workers) living in those areas and to compare human POPs levels with agricultural soil POPS concentrations and with food levels also.

However, some clarifications are necessary: please, read below and correct as required.

Lines 26-27: change “congeners” with “isomers” for HCHs and DDTs.

Line 72: remove “of”

Line 91: “has listed” and not “have listed”

Line 93: remove “f”

Line 114: complete the sentence “Several studies”

Line 115: the first “were” is “where”

Line 123: “collection”

Line 127: “represent” and not “represents”

Line 194: recoveries of HCHs weren’t fully satisfactory. Do you plan to improve them in the future? Does your lab normally participate in intercomparison excercises?

Line 196: PCBs

Line 230: after “components” put a dot

Line 235: I can’t read the range 0.01 to 22.200 mg∙kg−1 for OCPs (sum of HCHs and DDTs) in Table 2, why? Explain more.

Line 259: Do you have any idea why Aliyeva results were much lower than yours? Maybe sampling areas in Azerbaijan were different?

Line 277: put the same unit of measure (ng kg-1 or μg kg−1?)

Line 279 and Table 2: it is wrong reporting 4HCHs, because you have 3 HCHs (isomers) and HCB

Line 283: points on; in my opinion it would be better “points out on”

Lines 288-289: you wrote “The concentrations of PCB28, PCB52, PCB101, PCB118, PCB138, PCB153, PCB180 were below their detection limits.”. What do you mean? I think it isn’t correct. You ever quantified some congeners in all samples, or not?

Line 295: the UNECE-Europe Report from 2000 is not so new. Could you obtain a more recent report?

Line 298:

Line 338: replace “teir” with “their” or “the”

Tables: values need to be rounded off to significant figures so tables will be easier to read.

Author Response

Response to Reviewer 1 Comments

Point 1:

This paper deals with a very interesting topic, because it takes into consideration an important problem like residues of Persistent Organic Pollutants (POPs) in agricultural soils in Azerbaijan, a Country where in the past many tons of contaminants were used. I agree with the authors when they say that it is very important to know POPs amount and to residue it because of human and environmental health risks associated with their existence.

The paper is well structured, well done and easy to read. All necessary information is reported, so the paper is coherent.

This paper collected more information and it can be useful as the starting point for other studies as for example human biomonitoring studies. Do you know if human biomonitoring studies were carried out in the same regions where you picked up your samples? It would be very interesting to have POPs levels in women and men (also workers) living in those areas and to compare human POPs levels with agricultural soil POPS concentrations and with food levels also.

However, some clarifications are necessary: please, read below and correct as required.

Response to point 1: Authors would like to thank Reviewer 1 for his comments and suggestions, which very much contributed in improvement of the manuscript. All suggestions and recommendations have been adopted Corrections implemented within the text are listed below.

Point 2:

Lines 26-27: change “congeners” with “isomers” for HCHs and DDTs.

Response to point 2: The changes have been adopted to the manuscript.

Point 3:

Line 72: remove “of”

Response to point 3: The changes have been adopted to the manuscript

Point 4:

Line 91: “has listed” and not “have listed”

Response to point 4: The change has been adopted to the manuscript

Point 5:

Line 93: remove “f”

Response to point 5: The changes has been adopted to the manuscript

Point 6:

Line 114: complete the sentence “Several studies”

Response to point 6: The sentence has been completed.

Point 7:

Line 115: the first “were” is “where”

Response to point 7: The change has been adopted to the manuscript

Point 8:

Line 123: “collection”

Response to point 8: The changes have been adopted to the manuscript

Point 9:

Line 127: “represent” and not “represents”

Response to point 9: The changes have been adopted to the manuscript

Point 10:

Line 194: recoveries of HCHs weren’t fully satisfactory. Do you plan to improve them in the future? Does your lab normally participate in intercomparison excercises?

Response to point 10:The method described in the paper was previously validated for the very small level of HCHs occurring in Polish agricultural soils being <0.005 mg/kg therefore some parameters seemed not to be  satisfactory for so high level determined in the soils of Azerbaijan. The method was verified in the interlaboratory comparison and the obtained characteristics for HCHs met the criteria.

Point 11:

Line 196: PCBs

Response to point 11: The changes have been adopted to the manuscript

Point 12:

Line 230: after “components” put a dot

Response to point 12: The changes have been adopted to the manuscript

Point 13:

Line 235: I can’t read the range 0.01 to 22.200 mg∙kg−1 for OCPs (sum of HCHs and DDTs) in Table 2, why? Explain more.

Response to point 13: The mistake has been removed. The total concentration of OCP ranged  from 8.76 to 43.776 mg∙kg−1.

Point 14:

Line 259: Do you have any idea why Aliyeva results were much lower than yours? Maybe sampling areas in Azerbaijan were different?

Response to point 14: Yes, the research conducted by Aliyeva et al. (2012; 2013) were carried out in Azerbaijan but in other areas It indicates that other soils of Azerbaijan are not highly contaminated by POPs and our results may be treated as "hot spots" due to their pollution history.

Point 15:

Line 277: put the same unit of measure (ng kg-1 or μg kg−1?)

Response to point 15: The changes have been adopted to the manuscript

Point 16:

Line 279 and Table 2: it is wrong reporting 4HCHs, because you have 3 HCHs (isomers) and HCB

Response to point 16: The changes have been adopted to the manuscript. We totally removed the HCB value due to the insignificant contribution to the total HCHs concentration.

Point 17:

Line 283: points on; in my opinion it would be better “points out on”

Response to point 17: The changes have been adopted to the manuscript.

Point 18:

Lines 288-289: you wrote “The concentrations of PCB28, PCB52, PCB101, PCB118, PCB138, PCB153, PCB180 were below their detection limits.”. What do you mean? I think it isn’t correct. You ever quantified some congeners in all samples, or not?

Response to point 18: In the project (the founding source of these research), the scope of work included all described PCB analysis. Among these compounds, we detected PCB 138 and PCB 180 in all soil samples at the levels described in the table. 3. The content below the limit of detection (…and thus the possibility of determination these compounds by accessed techniques), was also a result which indicated that the analyzed soils was not contaminated by these compounds.

Point 19:

Line 295: the UNECE-Europe Report from 2000 is not so new. Could you obtain a more recent report?

Response to point 19: The new data has been added (from 2019 year) in lines 314-319 “According to  International Pollutants Elimination Network, the largest part of PCB deposition in Azerbaijan (reported in 2019) is associated with the unsecured storage of 83 tons of materials, including approximately 34.5 tons of PCBs oils, that need to be utilized [36]. As so far, no arrangements for storage of these materials were found, suggesting a lack of control over circulation of PCBs-containing equipment; and their possible emission into the environment. [36]

Point 20:

Line 338: replace “teir” with “their” or “the”

Response to point 20: The changes have been adopted to the manuscript.

Point 21:

Tables: values need to be rounded off to significant figures so tables will be easier to read.

Response to point 21: The changes have been adopted to the manuscript.

Reviewer 2 Report

Data on POPs residues in agricultural soils in the vicinity of historical locations of their storage and distribution in Azerbaijan presented in your study are very valuable and interesting and present a very good first step towards assessing the impact of „hot spots“ on surrounding agricultural soils.  However, collected data must be analyzed very carefully in order to avoid misinterpretations and wrong conclusions.

Firstly, you need to define and use the term „contaminated“ more precisely. For example, when discussing results obtained by Avazova (32) you stated that „The research revealed that 12 600 ha of agricultural areas out of a 247 studied 25 000 ha were contaminated with Σ6DDTs (n=50), at an average (range) concentrations of 248 0.573 (0.058–1.596) mg kg−1, and ΣHCHs (n=39) at concentrations of 0.028 (0.004–0.086) mg kg−1.“ But, if you refer to, for example, Dutch Target and Intervention Values for soil remediation, you will find that the target value for sum od DDTs is 0,01mg/kg and remediation value in 4 mg/kg and target and remediation values for the sum of HCHs are 0.01 and 2mg/kg respectively. Having that in mind, you should analyze more carefully whish soils are contaminated and which are only modified by pesticide presence.

Besides the data on the number of samples and sampling locations, you should present the data on how big is the area covered by sampling.

You did not discuss the ratio between DDT and its metabolites which could give you information on aerobic or anaerobic transformation processes (see M. Ricking and J. Schwarzbauer, Environ Chem Lett (2012) 10:317–323 and Chen et al. Pedosphere 25(6): 888–900, 2015)

Also, you did not analyze concentrations of HCH isomers which could give you valuable information on sources of contamination (see and Chen et al. Pedosphere 25(6): 888–900, 2015)

For soil classification regarding PAH content, you can refer to Maliszewska-Kordybach, B., 1996. Polycyclic aromatic hydrocarbons in agricultural soils in Poland: preliminary proposals for criteria to evaluate the level of soil contamination. Appl. Geochem. 11, 121-127, where soils are divided in the following groups regarding total PAHs content: not contaminated (total PAHs < 200 μg/kg), weakly contaminated (total PAHs is 200-600 μg/kg), contaminated (total PAHs is 600-1000 μg/kg) and heavily contaminated (total PAHs is >1000 μg/kg).

I would strongly recommend you to reanalyze your data set and rewrite the Results and Discussion section.

Author Response

Response to Reviewer 2 Comments

Point 1:

Data on POPs residues in agricultural soils in the vicinity of historical locations of their storage and distribution in Azerbaijan presented in your study are very valuable and interesting and present a very good first step towards assessing the impact of „hot spots“ on surrounding agricultural soils.  However, collected data must be analyzed very carefully in order to avoid misinterpretations and wrong conclusions.

Response to point 1: Authors would like to thank Reviewer 2 for his/her comments and suggestions, which contributed very much to improve the manuscript. All suggestions and recommendations have been adopted, and changes to the text are listed below.

Point 2

Firstly, you need to define and use the term „contaminated“ more precisely. For example, when discussing results obtained by Avazova (32) you stated that „The research revealed that 12 600 ha of agricultural areas out of a 247 studied 25 000 ha were contaminated with Σ6DDTs (n=50), at an average (range) concentrations of 248 0.573 (0.058–1.596) mg kg−1, and ΣHCHs (n=39) at concentrations of 0.028 (0.004–0.086) mg kg−1.“ But, if you refer to, for example, Dutch Target and Intervention Values for soil remediation, you will find that the target value for sum od DDTs is 0,01mg/kg and remediation value in 4 mg/kg and target and remediation values for the sum of HCHs are 0.01 and 2mg/kg respectively. Having that in mind, you should analyze more carefully whish soils are contaminated and which are only modified by pesticide presence.

Response to point 2: We agree with the Reviewer 2 opinion, that the manuscript does not specified term „soil contamination” which may cause some confusion. Nevertheless, any national regulations have been adopted in Azerbaijan that show POPs limit concentration values for agricultural soils (contrary to European countries or United States). Moreover, the regulations implemented in various countries are not uniform and differ substantially which leads to many controversial issues about contaminated soils. Therefore, in our paper, we have not adopted any limit values for the analyzed DDT, HCH and PAH compounds because it would be difficult to choose specific regulations (which one? and why?) applicable in these specific conditions. According to the Reviewer 2 suggestion, we corrected the statement "contaminated soils" in the revised manuscript.

Point 3

Besides the data on the number of samples and sampling locations, you should present the data on how big is the area covered by sampling.

Response to point 3: The relevant information has been listed in Table 1.

Point 4

You did not discuss the ratio between DDT and its metabolites which could give you information on aerobic or anaerobic transformation processes (see M. Ricking and J. Schwarzbauer, Environ Chem Lett (2012) 10:317–323 and Chen et al. Pedosphere 25(6): 888–900, 2015)

Response to point 4: Appropriate changes have been introduced to the manuscript (lines 275-289): ”Under aerobic conditions, DDT is degraded to DDE while under  anaerobic circumstances DDD predominates. High percentage of 4,4′-DDE confirms on microbial degradation in presence of oxygen in soils. The ratio of 4,4′-DDT/(4,4'-DDE + 4,4′-DDD) is an indicator whether 4,4′-DDT in soils is degraded or inputted recently [27,50].  In the present study, the proportion of 4,4′-DDE/ 4,4'DDD accounted for comparatively from 0.9 to 6.46 and the ratio of 4,4′-DDT/(4,4′- DDE + 4,4′-DDD) ranged from 0.41 to 61.92 with mean value of 8.76. The results suggested that the DDT residues are mainly subjected to anaerobic degradation in most sampling sites, but generally high levels of DDTs concentration can affect difficult degradation of these compounds. Moreover, the 2,4′-DDT/4,4′-DDT ratio can be used to distinguish DDT sources [50]. The 2,4′-DDT/4,4′-DDT ratio varies from 0.02 to 12,42 with mean value of 0,27 indicating that the DDT residues derived from technical DDT input and dicofol. The 2,4′-DDT/4,4′-DDT ratio for 48% of samples exhibited value above 0.3, implying the dominant contribution of  mixed sources of DDT [50]”.

Point 5

Also, you did not analyze concentrations of HCH isomers which could give you valuable information on sources of contamination (see and Chen et al. Pedosphere 25(6): 888–900, 2015)

Response to point 5: Appropriate changes have been introduced to the manuscript (lines 289-304): “The dominant compound was α-HCH, which contributed about 73 % of the Σ3HCHs, followed by β- HCH and γ-HCH (Table 2). The isomer ratio of α-HCH to γ-HCH for most samples significantly exceeded 1.0, which confirms the high stability of the α isomer, while γ-HCH has been subjected to fast biological and chemical degradation and indicate ability to leaching because of its relatively high water solubility [19,35,7]. The α-HCH and γ-HCH can both be converted to β-HCH, which exhibited the lowest degradation rate and vapor pressure [6,35,50]. In our study, the proportion between this isomers (β-HCH/( α-HCH+ γ-HCH)) were from 0 to 13.66 with median value of 2.10, which points out on a historical nature of these contaminants in soils or its deposition caused by long-range transport [6,50].

The differences in composition of HCH isomers in the environment could indicate different sources of contamination. Technical HCHs contain high concentrations of α-HCH (60%–70%), compared to β-HCH (5%–12%) and γ-HCH (10%–12%). Therefore, despite the application of technical HCHs which has been forbidden in many years ago, its long degradation time may cause the high persistence of these contaminants in soil [50]”.

 Point 6

For soil classification regarding PAH content, you can refer to Maliszewska-Kordybach, B., 1996. Polycyclic aromatic hydrocarbons in agricultural soils in Poland: preliminary proposals for criteria to evaluate the level of soil contamination. Appl. Geochem. 11, 121-127, where soils are divided in the following groups regarding total PAHs content: not contaminated (total PAHs < 200 μg/kg), weakly contaminated (total PAHs is 200-600 μg/kg), contaminated (total PAHs is 600-1000 μg/kg) and heavily contaminated (total PAHs is >1000 μg/kg).

Response to point 6: Appropriate changes have been introduced to the manuscript (lines: 352-360): “According to the criteria described in the publication of Maliszewska-Kordybach [51] the most of the soils analyzed in our study can be classified as non-contaminated (10% of samples) and weakly contaminated (52% of samples) because they were characterized by the ∑16PAHs concentration below 200 µg kg-1 and 200 - 600 µg kg-1, respectively. The rest of the soils can be recognized as contaminated (10% of samples with PAHs concentration between 600 and 1000 µg kg-1) and heavily contaminated (29% of samples with PAHs concentration below 1000 µg kg-1). Maliszewska-Kordybach [51] proposed four classes of soil contamination by PAH. The threshold values express the absolute sum of the 16 PAH concentration in the soil, disregarded PAH composition and soil properties. The limit values at the level 200, 600 and 1000 µg kg-1 have been derived from the results of determinations of PAH content in European soils [51].”

Point 7

I would strongly recommend you to reanalyze your data set and rewrite the Results and Discussion section.

Response to point 7: We re-analyzed our results and complemented some paragraphs in the discussion (lines 235, 244, 248-255, 260-262, 275-288, 289-302, 352-360).

Round 2

Reviewer 2 Report

The Results and Discussion section is improved and the Manuscript can be accepted in the present form.